# Host-Pathogen Interactions in *K. pneumoniae* Urinary Tract Infections: Investigating Genetic Risk Factors in the Taiwanese Population

**DOI:** 10.3390/diagnostics14040415

**Published:** 2024-02-14

**Authors:** Chi-Sheng Chen, Kuo-Sheng Hung, Ming-Jr Jian, Hsing-Yi Chung, Chih-Kai Chang, Cherng-Lih Perng, Hsiang-Cheng Chen, Feng-Yee Chang, Chih-Hung Wang, Yi-Jen Hung, Hung-Sheng Shang

**Affiliations:** 1Division of Clinical Pathology, Department of Pathology, Tri-Service General Hospital, National Defense Medical Center, Taipei 114, Taiwancindyft12@gmail.com (H.-Y.C.);; 2Center for Precision Medicine and Genomics, Tri-Service General Hospital, National Defense Medical Center, Taipei 114, Taiwan; 3Division of Rheumatology/Immunology and Allergy, Department of Medicine, Tri-Service General Hospital, National Defense Medical Center, Taipei 114, Taiwan; hccheng@ndmctsgh.edu.tw; 4Division of Infectious Diseases and Tropical Medicine, Department of Medicine, Tri-Service General Hospital, National Defense Medical Center, Taipei 114, Taiwan; fychang@mail.ndmctsgh.edu.tw; 5Department of Otolaryngology-Head and Neck Surgery, Tri-Service General Hospital, National Defense Medical Center, Taipei 114, Taiwan; 6Division of Endocrinology and Metabolism, Department of Internal Medicine, Tri-Service General Hospital, National Defense Medical Center, Taipei 114, Taiwan

**Keywords:** *Klebsiella pneumoniae*, urinary tract infections, genome-wide association studies

## Abstract

Background: *Klebsiella pneumoniae* (*K. pneumoniae*) urinary tract infections pose a significant challenge in Taiwan. The significance of this issue arises because of the growing concerns about the antibiotic resistance of *K. pneumoniae*. Therefore, this study aimed to uncover potential genomic risk factors in Taiwanese patients with *K. pneumoniae* urinary tract infections through genome-wide association studies (GWAS). Methods: Genotyping data are obtained from participants with a history of urinary tract infections enrolled at the Tri-Service General Hospital as part of the Taiwan Precision Medicine Initiative (TPMI). A case-control study employing GWAS is designed to detect potential susceptibility single-nucleotide polymorphisms (SNPs) in patients with *K. pneumoniae*-related urinary tract infections. The associated genes are determined using a genome browser, and their expression profiles are validated via the GTEx database. The GO, Reactome, DisGeNET, and MalaCards databases are also consulted to determine further connections between biological functions, molecular pathways, and associated diseases between these genes. Results: The results identified 11 genetic variants with higher odds ratios compared to controls. These variants are implicated in processes such as adhesion, protein depolymerization, Ca^2+^-activated potassium channels, SUMOylation, and protein ubiquitination, which could potentially influence the host immune response. Conclusions: This study implies that certain risk variants may be linked to *K. pneumoniae* infections by affecting diverse molecular functions that can potentially impact host immunity. Additional research and follow-up studies are necessary to elucidate the influence of these risk variants on infectious diseases and develop targeted interventions for mitigating the spread of *K. pneumoniae* urinary tract infections.

## 1. Introduction

Urinary tract infection (UTI) is the most common bacterial infection worldwide. It is also an important issue in hospitalized patients because of increased antibiotic resistance and increased morbidity and mortality in immunocompromised patients or those with cancer [1]. In both ambulatory and hospitalized patients, *K. pneumonia* is the second most common bacterial pathogen associated with UTIs around the world, and the frequency of *K. pneumoniae* infection has increased in hospitals or nursing homes [2]. Although *K. pneumonia* is considered an opportunistic pathogen, several specific capsular serotypes of *K. pneumonia* with increased production of capsule polysaccharide are considered to be hypervirulent *K. pneumonia*. In addition to the increased production of capsule polysaccharide, which helps the bacteria resist phagocytosis, it is also believed that these hypervirulent *K. pneumonia* may have an impact on host immune reactions and cause invasive infections [3]. They are also capable of hydrolyzing several kinds of antibiotics, especially carbapenems, which are so-called *Klebsiella pneumoniae* carbapenemase-producers, making them a threat to vulnerable patients due to their resistance to multiple antibiotics [4] and causing considerable healthcare costs [2].

To further understand the possible risk factors between *K. pneumonia* and the hosts, we use genome-wide association studies (GWAS) in order to determine possible common factors in the genomes of patients with UTIs caused by *K. pneumonia*.

## 2. Materials and Methods

### 2.1. Study Participants and Ethical Approval

All participants in this study were recruited from the Tri-Service General Hospital (Taipei, Taiwan; TSGH) to join the Taiwan Precision Medicine Initiative (TPMI) [5]. The TPMI is held by Academia Sinica (Taipei, Taiwan) in partnership with 16 top medical centers in Taiwan and aims to establish a database consisting of comprehensive clinical data and the genetic profiles of one million Taiwanese Han population participants. Participants were recruited from medical centers and genotyped using Academia Sinica. The protocol of this study was reviewed and approved by the Institutional Review Board of the Tri-Service General Hospital (NO.: 2-108-05-038).

The profiles of all patients included in our study are detailed in Table 1. Additionally, we have listed several underlying diseases that are considered to pose a vulnerability to UTI, including diabetes mellitus, malignancy, chronic kidney disease, and a history of urinary tract stones. These medical histories were documented at the time of enrollment in the TPMI project for each patient.

### 2.2. Pathogen Identification in a Patient with UTI

Urine samples for bacterial culture were collected at the same time that the patients were transferred to our hospital because of UTI-related symptoms. All urine culture results were confirmed by MALDI-TOF with VITEK^®^ MS (bioMérieux, Marcy-l’Étoile, France).

### 2.3. Genotyping

Genotyping of TPMI participants was performed as follows: First, approximately 3 mL of peripheral blood per participant was collected into EDTA vacutainers. Genomic DNA was extracted from blood cells using the QIAsymphonyTM SP Stander protocol (Qiagen, Hilden, Germany). Next, the genomic DNA was characterized by its variants under a customized SNP array called the Axiom Genome-Wide TPM plate, which was developed by Academia Sinica (Taipei, Taiwan) and Thermo Fisher Scientific Inc. (Waltham, MA, USA).

### 2.4. GWAS Analysis

The steps of the GWAS are presented in Figure 1. A total of 1860 patients with UTI were selected from among the TSGH TPMI participants. After removing some unqualified samples, 1825 patients, including 62 UTIs with *K. pneumoniae*, were defined as case groups. The remaining 1763 patients with UTIs infected with other pathogens, such as *E. coli*, were included in the control group (Table 1). Genotyping data of these UTI patients from TPM array results were first filtered out by low typing call rate SNP (<80%) and then applied to association analysis by chi-squared test (case group vs. control) using PLINK 1.9 (https://zzz.bwh.harvard.edu/plink/ (accessed on 16 January 2023)) software [6]. Variants with low quality (minor allele frequency less than 0.05 and Hardy-Weinberg equilibrium less than 1 × 10^−6^) were removed using PLINK, and highly significant *p*-values (less than 1 × 10^−5^) were selected for further analysis. For linkage disequilibrium (LD) analysis, the GWAS results were loaded onto LocusZoom [7] to observe the LD relationship of each variant.

### 2.5. Variant Annotations

For variant annotations, genes were identified using the RefSeq Database (https://www.ncbi.nlm.nih.gov/refseq/ (accessed on 16 January 2023)) based on wANNOVAR (https://wannovar.wglab.org/ (accessed on 16 January 2023)) [8], and to compare allele frequency in other race populations, the public domain databases 1000 genome [9], Genome Aggregation Database (gnomAD) [10], and Taiwan BioBank (https://taiwanview.twbiobank.org.tw/index (accessed on 17 January 2023)) were applied, respectively. Gene functional characterization was performed using Gene Ontology [11], and related pathways were searched in the Reactome pathway database [12] using Enrichr (https://maayanlab.cloud/Enrichr/ (accessed on 17 January 2023)) [13]. To investigate the variant expression profiles, the Protein Atlas (https://www.proteinatlas.org/ (accessed on 17 January 2023)) and GTEx (https://gtexportal.org/home/ (accessed on 17 January 2023)) were used. In addition, DisGeNET (https://www.disgenet.org/ (accessed on 18 January 2023)) and MalaCards (https://www.malacards.org/ (accessed on 18 January 2023)) [14] were used to determine relationships between genes and diseases.

## 3. Results

### 3.1. Variants from GWAS Analysis

After GWAS analysis, the filtered case and control groups (case group: 62 patients; and control group: 1763 patients), together with 241,217 SNP, passed through the SNP calling rate, minor allele frequency, and Hardy-Weinberg equilibrium. There were 13 significant variants obtained from the case group, and the Manhattan plot of the GWAS results is presented in Figure 2. It was found that variants from the case group had a higher odds ratio than the control groups (Table 2), and most SNP allele frequencies (rs10411896, rs11672710, rs12313615, rs61875193, rs62126347, rs62126348, rs76541491, and rs117166327) were similar to the East Asian and Taiwan Han (TPMI and Taiwan Biobank) populations, which were different from other human races (Table 3). 

According to the RefSeq results, these risk alleles belonged to the genes C12orf75, CASC18, HSPBP1, IQSEC1, KCNN3, MAGEC2, MICAL2, NUP210, PTCHD1-AS, SPANXN4, SUSD5, and TEX14 in the case group (Table 2). Interestingly, MIR4435-2HG and PTCHD1-AS are non-coding RNAs that regulate target genes without coding proteins. 

Further analysis using linkage disequilibrium (LD) showed that in the case group, the variant rs61875193 had a high correlation with rs58910113 (Figure 3A), and rs73387413, which is highly associated with rs12313615, also showed the same result (Figure 3B). Surprisingly, rs11672710 not only had a high association with rs62126347, rs62126348, and rs10411896 in HSPBP1 but also with rs4806651 and rs4806653 in PPP6R1 and rs4337407 in TMEM86B (Figure 3B). 

### 3.2. Functional Annotations of Risk Genes

Table 4 shows that 35 significant GO terms are identified, and further categories in Figure 4A present 14 classes, including carbohydrate catabolism, cation-potassium transport, cell-cell interaction, heart development, keratinocyte migration, lipid metabolism, mitotic phase, ncRNA transport, protein depolymerization, protein phosphoprotein, protein ubiquitination, purine metabolism, RNA catabolism, and tRNA catabolism. These biological functions belong to cell growth, metabolism, migration, and interactions. Notably, there are several genes involved in Ca^2+^/K^+^ regulation (*KCNN3*), protein ubiquitination (*HSPBP1* and *MAGEC2*), and the breaking down of protein polymers (*MICAL2*), which might have connections to the *K. pneumonia* infections. *TMEM86B* and *PPP6R1*, which were identified from LD analysis, played a role in lipid metabolism and phosphatase activity, respectively, which may affect immune cell activity and cause *K. pneumoniae* infection.

### 3.3. Pathway Analysis

After searching the Reactome database, 26 significant pathways were successfully identified (Table 5), and 11 classes of pathways are shown in Figure 4B: cation-potassium transport, MET, mRNA metabolism, mRNA transport, NS1/NS2 protein, nuclear pore complex, phosphatidylcholine acyl chain, Rev/Vpr protein, ribonucleoproteins, SUMOylation, and TPR in papillary thyroid carcinoma. Many pathways are involved in mRNA regulation and SUMoylation by the gene *NUP210*, which may alter the immune system relative to the UTI caused by *K. pneumoniae*. Pathways identified from *KCNN3* (Ca^2+^-activated K^+^ channels) and *TMEM86B* (acyl chain remodeling of phosphatidylcholine) fit the GO results, indicating that patients with abnormal cation-potassium transport or abnormal lipid metabolism are at a high risk of *K. pneumoniae* infection.

### 3.4. Investigation of Risk Gene Expressions

According to the Protein Atlas results in Table 6, the RNA expression of the genes *C12orf75*, *FAM209B*, *HSPBP1*, *IQSEC1*, *KCNN3*, *LSM3*, *MICAL2*, *NUP210*, *PPP6R1*, *SOCAR*, *SUSD5*, *TMEM86B*, and *TNS3* was detected in immune cells. In particular, some of these proteins, such as *C12orf75*, *FAM209B*, *KCNN3*, *MICAL2*, *PPP6R1*, and *TNS3*, showed high immune cell specificity. Combined with the GO and Rectome analysis results, the changes in the variants further regulated the activity of immune cells. Unfortunately, few expression patterns were recorded in the GTEx database for these variants; only rs140411896 in *HSPBP1* and rs4337407 in *TMEM86B* presented changes in gene expression in whole blood compared to the non-mutated alleles (Figure 5A: *HSPBP1* down expression occurred with the rs10411896 mutation and Figure 5B: *TMEM86B* up expression with the rs4337407 mutation). These findings are worthy of further study to determine the relationship between the Taiwan Han population’s special genetic allele affections in immune cells and *K. pneumoniae* infection in the urinary tract.

## 4. Discussion

### 4.1. Disease Association Analysis

*K. pneumonia* is a Gram-negative bacterium commonly found in the human gut and environment, and it is also known to be the major cause of healthcare-associated infections, especially pneumonia, bloodstream infections, and urinary tract infections. Previous studies have reported that *K. pneumoniae* infection is a rare cause of community-acquired pneumonia (CAP) in North America, Europe, and Australia [15]. However, investigations in eight Asian countries, including Taiwan, have reported that *K. pneumoniae* is highly prevalent [16]. Several drug-resistant *K. pneumoniae* variants have also posed a significant public health threat due to their difficulty in treatment and higher mortality rates, such as carbapenem-resistant *K. pneumonia* (CRKP), extended-spectrum beta-lactamase (ESBL)-producing *K. pneumonia*, and hypervirulent *K. pneumoniae* (hvKp). Due to the widespread resistance to antibiotics and the possibility of causing more severe infections, it is crucial to understand the specific resistance patterns and infectious pathogenesis, which can contribute to improving disease prevention, infection control, and further antibiotic development.

Patients with chronic diseases such as diabetes, cancer, and chronic kidney disease are believed to have a higher risk of developing *K. pneumoniae* UTIs [17]. Although our research revealed no significant differences in the ratio of underlying conditions and diseases between the case group and the control group, which included diabetes mellitus, malignancy, chronic kidney disease, and other chronic viral infections, we did observe significant differences in certain genetic variants. Consequently, it suggests that the variation leading to the infection pathogen of urinary tract infections (UTI) in these patients might be more closely related to their genes than their underlying conditions.

Nevertheless, all the chronic diseases mentioned above have a prolonged and enduring course. Predicting whether patients in both groups will develop these diseases later on remains challenging. Nonetheless, this information could still provide valuable insights into the potential pathogenesis of UTI with *K. pneumoniae* infection.

### 4.2. Genotyping

By October 2022, the TPMI team had 382,259 genotyped samples. This genomic information is also combined with other clinical records, such as ICD10 disease codes, laboratory tests, medications (drug usage), vital signs, image descriptions, and operation notes, which makes the TPMI the largest genetic health association study compared to other similar projects held in the United States or Japan. The TPMI SNP array was modified from the Axiom Genome-Wide SNP Array Plate and can test approximately 130 thousand known risk variants, 580 thousand mapping SNPs, and 20 thousand copy number variant markers based on Taiwanese reference genomes and Taiwan Biobank whole genome sequencing data. 

The risk variants from the GWAS in this study showed that most allele frequencies in the East Asian population, which were also close to the Taiwanese population, were the highest compared to other species. This phenomenon was also observed in the variants rs4337407, rs4806651, rs4806653, and rs58910113 from the LD analysis, not only showing a higher LD association with lead variants than with species, but also that their minor allele frequencies were higher in African or Asian population groups. Some variants of rs530922, rs1034726, rs6678353, rs12687449, and rs117166327 show non-or rare frequency records in the published database, which may be Taiwan-specific variants for *K. pneumoniae* infection risk factors.

*K. pneumoniae* is an opportunistic pathogen with compromised immune systems due to phagocytosis by epithelial cells, macrophages, neutrophils, and DCs, or weakened by other infections [4]. As shown in Table 6 and Figure 5, the genes *C12orf75*, *FAM209B*, *HSPBP1*, *IQSEC1*, *KCNN3*, *LSM3*, *MAGEC2*, *MICAL2*, *NUP210*, *PPP6R1*, *SOCAR*, *SPANXN4*, *SUSD5*, *TMEM86B*, and *TNS3* were involved in immune cells, and GTEx analysis showed that *HSPBP1* and *TMEM86B* expression could be altered by specific genetic variants. Thus, our findings can also contribute to further clarifying the role of these risk genes in the relationship between *K. pneumoniae* infection and immune system defects.

Galeas-Pena reported that *K. pneumoniae* infection activates the signaling of nuclear factor kappa B (NF-κβ), promoting the recruitment of immune cells [18]. The gene *PPP6R1*, which plays an important role in protein phosphorylation, has been shown to limit the activation of the NF-κβ pathway by reducing IκBε phosphorylation [19]. This mechanism is probably caused by Slfn2 interacting with *PPP6R1*, leading to reduced type I IFN-induced activation of NF-κB signaling [20]. *PPP6R1* is also known as a regulatory subunit of PP6, which can negatively regulate NF-κβ signaling [21], highlighting the importance of phosphatase activity in the immune activation process during *K. pneumoniae* infection. 

Bacterial cell-cell surface interactions are the core source of pathogen infection. Previous reports have demonstrated that alterations in focal adhesion and the actin cytoskeleton play important roles in the bacterial invasion of host cells [22]. In addition, Hsu et al. indicated that Rho is involved in the activation of focal adhesion through the phosphorylation of focal adhesion kinase, which could affect the induction of cell cytoskeleton rearrangements. The same research also indicated that Cdc42 and the PI3K/Akt pathway are activated to induce cell cytoskeleton rearrangement via *K. pneumoniae* adhesion [23]. Functional annotation of *IQSEC1* and *MICAL2*, including focal adhesion and actin filament depolymerization, showed that mutations in both genes could pose potential risks to *K. pneumoniae* invasion.

In addition to the innate immune perturbations associated with *K. pneumoniae*, dysregulation of electrolyte homeostasis [24] and protein ubiquitination may also affect host immune immunity. Immune responses activated by ion channel transporters such as calcium, magnesium, sodium, potassium, and zinc have been reported, and pathogen infection also requires ion equilibrium changes in the living cell environment [25]. According to the results of studying Galleria mellonella, which is used as a *K. pneumoniae* infection model organism, the amounts of calcium, potassium, magnesium, and phosphorus are altered during *K. pneumoniae* infection [26]. Zhang also indicated that the calcium signaling pathway increased macrophage activity resulting from *K. pneumoniae* infection [27], and further research presented that *TRPC1* (Transient receptor potential channel 1) could mediate Ca^2+^ entry and activate NF-κβ/Jun, leading to the proinflammatory response [28]. Thus, the mutation of *KCNN3*, which belongs to the Ca^2+^-activated potassium channels, may increase Ca^2+^ accumulation in living cells through *TRPC1* and promote Ca^2+^ levels during *K. pneumoniae* infection. 

Research has shown that *K. pneumonia* can interfere with or suppress host immunity and inflammation reactions by affecting the ubiquitination of various cellular signals [29,30]. What is more, a small ubiquitin-related modifier (SUMO) can be involved in various biological processes, including immunopathology and inflammation [31], which can also be affected by *K. pneumonia* and cause extensive inflammation damage [32]. In conclusion, according to these results, protein ubiquitination plays a role in the activation of the innate immune response and subsequent release of cytokines when facing bacterial invasion. Furthermore, our finding of risk genes for ubiquitination and SUMOylation may play a great part in the cellular immune responses against *K. pneumoniae* invasion. Thus, patients with risk genes affecting the regulation of the innate immune response may be more susceptible to *K. pneumonia* infection, although further research is needed to identify the relationships between these factors and *K. pneumoniae* infection, especially UTIs.

### 4.3. Limitations

Despite all the findings mentioned above, there are still some limitations to our research. Although the results had reached statistical significance, there still might be some pitfalls for our results to be extrapolated to the normal population due to the low number of case-group specimens. Our research excluded several samples that had co-infections with other pathogens in order to demonstrate our findings more clearly. However, it might have further information that can contribute to the pathogenesis. 

Moreover, UTI has been known to have a high recurrence rate [2], but it is difficult to compare the recurrent rate and the prognosis between the case group and control group because it is difficult to define the first episode of UTI and track the following medical history of recurrent UTI events. We still need further delicate studies to reveal the difference between these patients. However, we believe our findings can pave the way for the following research.

## 5. Conclusions

The detection of a GWAS based on the Taiwanese population in this study supports the hypothesis that some risk variants might correspond to *K. pneumoniae* UTI. In addition, these variants probably affect diverse bio-molecular functions, such as Ca^2+^-activated potassium channels, focal adhesion, NF-κβ signaling, SUMOylation, and protein ubiquitination. These functional genes may affect the host immune system, thereby increasing the ability of *K. pneumoniae* to invade. Thus, our results might also explain why, although *K. pneumoniae* UTI is believed to be related to healthcare, it is still the second most common pathogen in uncomplicated UTI. Overall, we still need further studies of the detailed molecular mechanisms between the host immune system and *K. pneumonia* UTI to fully understand the pathogenesis of the *K. pneumoniae* infection.

## Figures and Tables

**Figure 1 diagnostics-14-00415-f001:**
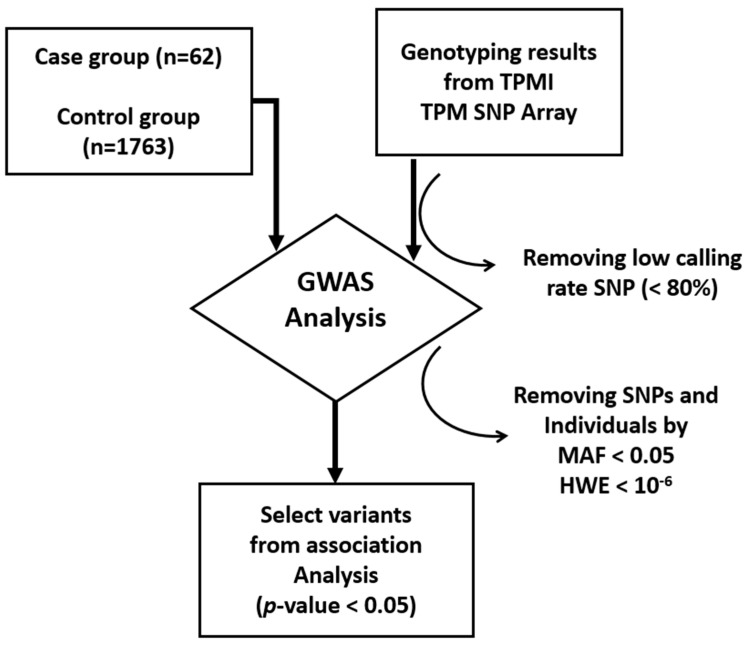
GWAS analysis pipeline: Case groups (UTI with *K. pneumonia* only) and control groups (UTI with other pathogens) and genotyping identified by the TPMI project were loaded into PLINK and used the chi-squared test for detecting risk factors. High-significance variants were selected according to a *p*-value < 0.05.

**Figure 2 diagnostics-14-00415-f002:**
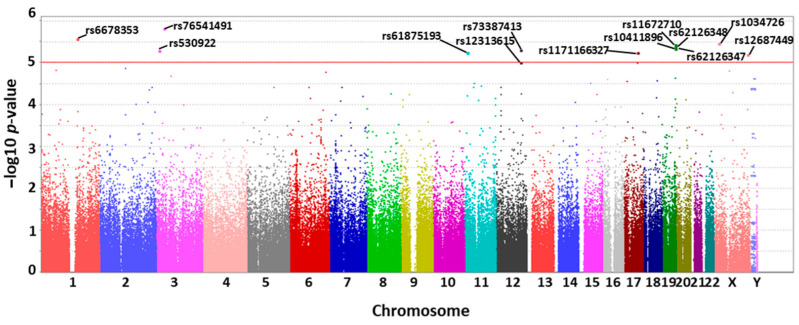
Manhattan plot of GWAS results in patients with UTI by identifying bacterial species through culture: UTI cases were selected from the Taiwan Han Genetic Project by a customized genotyping array composed of 241,217 SNPs. When comparing 62 UTI patients infected by *K. pneumonia* only and 1763 with other pathogens (e.g., *E. coli*), there are 13 different variant positions with unique SNP IDs (up the red line in the plot) with high significance (*p*-value < 10^−5^). These SNP IDs are characterized in chromosomes 1 (rs6678353), 3 (rs530922 and rs76541491), 11 (rs61875193 and rs61875193), 12 (rs73387413 and 12313615), 17 (rs1171166327), 19 (rs11672710, rs62126348, rs62126347, and rs10411896), and X (rs1034726 and rs12687449).

**Figure 3 diagnostics-14-00415-f003:**
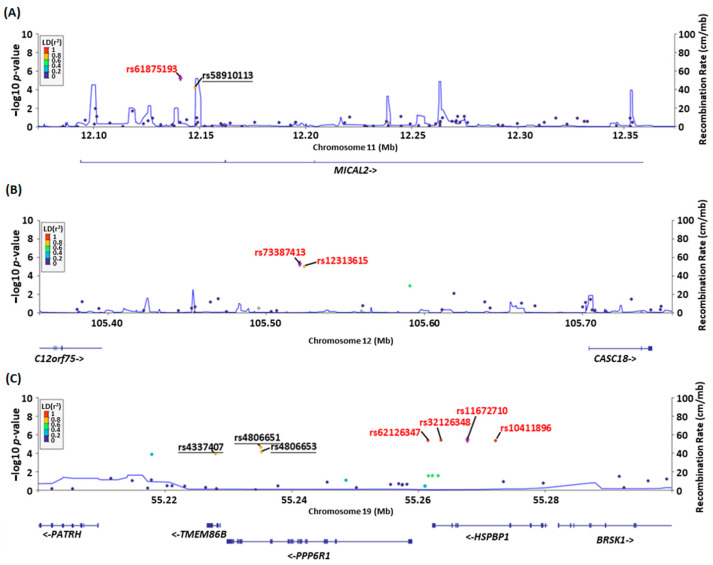
Linkage disequilibrium (LD) relationship of variants in the case group: Using the lower variant as the leading SNP (violet diamond) and East Asian population frequency, we detected (**A**) rs61875193 and rs58910113 in chr11 and (**B**) rs73387413 and rs12313615 in chr12. (**C**) rs62126347, rs62126348, rs11672710, and rs10411896 have a high LD r square correlation. Moreover, rs4337407 and rs4806651 in gene PPP6R1 and rs4806653 in gene TMEM86B also have a high LD association with rs11672710. Think and bold lines are intron and exon regions; the arrows mean transcription direction (left to right: positive stand; right to left: negative strand).

**Figure 4 diagnostics-14-00415-f004:**
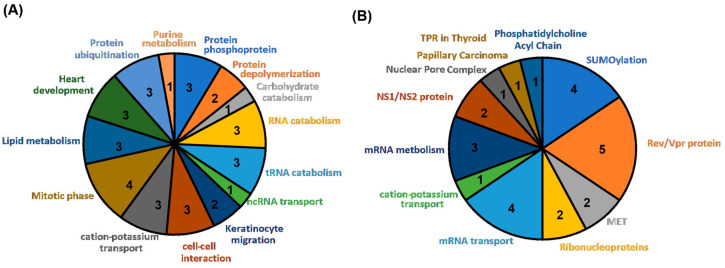
Summary of GO (**A**) and Reactome pathway (**B**) annotation: Variant genes were uploaded to the Gene Ontology and Reactome database, and the identified GO and Reactome terms (*p*-value < 0.05) were collected. Similar functional terms were pooled together in the pie chart.

**Figure 5 diagnostics-14-00415-f005:**
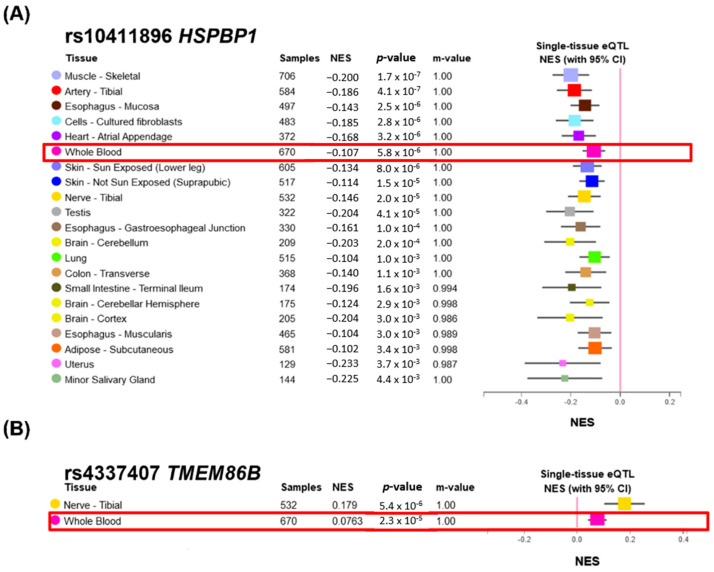
GTEx tissue eQTLs of variants in HPSPBP (**A**) and TMEM86B (**B**): All tissue eQTLs were selected based on a *p*-value < 0.01, and normalized effect size (NES) is defined as the effect of the alt allele relative to the ref allele in the human genome reference by computing in a normalized space where magnitude has no direct biological interpretation. The m-value means were made by using METASOFT to identify the posterior probability, and when the m-value is ≥0.9, it indicates that the tissue is predicted to have an eQTL effect.

**Table 1 diagnostics-14-00415-t001:** Sample group information for GWAS analysis.

	Case Group (*n* = 62)	Control Group (*n* = 1763)	*p* Value
*K. pneumonia*	62	0	-
Other pathogens	0	1763	-
Sex			
Male, no, (%)	18 (29.0%)	394 (22.3%)	0.216
Female, no, (%)	44 (71.0%)	1369 (77.7%)	
Age, median (SD), years	62.5 ± 16.52	61 ± 18.45	
<20, no (%)	-	15 (0.9%)	0.466
20–30, no, (%)	3 (4.8%)	161 (9.1%)	0.245
31–50, no, (%)	9 (14.5%)	387 (22.0%)	0.163
51–70, no, (%)	29 (46.8%)	710 (40.3%)	0.305
>70, no, (%)	21 (33.9%)	490 (27.8%)	0.295
Medical Hx			
w/o * any Hx, no, (%)	16 (25.8%)	444 (25.2%)	0.912
Diabetes mellitus, no, (%)	24 (38.7%)	659 (37.4%)	0.832
Malignancy, no, (%)	7 (11.3%)	195 (11.1%)	0.955
Chronic kidney disease, *n*, (%)	8 (12.9%)	208 (11.8%)	0.791
Autoimmune disease, no, (%)	6 (9.7%)	165 (9.4%)	0.933
Urinary tract stone, no, (%)	5 (8.1%)	139 (7.9%)	0.959
Chronic virus infection, no, (%)	5 (8.1%)	137 (7.8%)	0.932

* w/o: without.

**Table 2 diagnostics-14-00415-t002:** Variants obtained from the GWAS result with high significance in the case group.

CHR	Position	SNP ID	Ref ^a^	Alt ^b^	*p*-Value ^c^	Odd Ratio	Region	Relative Gene
1	154728516	rs6678353	G	T	2.69 × 10^−6^	2.44	intronic	*KCNN3*
3	13236137	rs530922	G	A	5.21 × 10^−6^	2.41	intergenic	*IQSEC1*; *NUP210*
3	33156385	rs76541491	C	T	1.53 × 10^−6^	2.76	intronic	*SUSD5*
11	12141089	rs61875193	T	C	5.84 × 10^−6^	2.75	intronic	*MICAL2*
12	105521299	rs73387413	G	A	4.98 × 10^−6^	2.52	intergenic	*C12orf75*; *CASC18*
12	105524194	rs12313615	G	A	9.68 × 10^−6^	2.38	intergenic	*C12orf75*; *CASC18*
17	58637506	rs117166327	G	A	5.64 × 10^−6^	2.89	intronic	*TEX14*
19	55261525	rs62126347	G	A	4.39 × 10^−6^	2.87	downstream	*HSPBP1*
19	55263581	rs62126348	T	C	4.06 × 10^−6^	2.88	intronic	*HSPBP1*
19	55267754	rs11672710	C	T	3.70 × 10^−6^	2.90	intronic	*HSPBP1*
19	55272184	rs10411896	A	C	4.57 × 10^−6^	2.87	intronic	*HSPBP1*
X	22810351	rs1034726	G	C	3.51 × 10^−6^	3.12	ncRNA_intronic	*PTCHD1-AS*
X	142786494	rs12687449	C	T	6.42 × 10^−6^	2.83	intergenic	*MAGEC2*; *SPANXN4*

^a^ Allele from the control group. ^b^ Allele from the case group. ^c^ Filtered by *p* < 0.05.

**Table 3 diagnostics-14-00415-t003:** Variant allele frequency from other genetic projects.

This Study				TPMI ^a^	Twbank ^b^	1000 g ^c^				gnomAD ^d^				
SNP ID	Alt	Case	Control			AFR ^e^	AMR	EAS	EUR	AFR	AMR	EAS	FIN	NFE
rs6678353	T	0.331	0.168	0.188	0.179	-	-	-	-	-	-	-	-	-
rs530922	A	0.702	0.494	0.493	0.489	-	-	-	-	-	-	-	-	-
rs76541491	T	0.234	0.100	0.101	0.110	0.001	0.016	0.120	0.015	0.005	0.004	0.103	0.022	0.021
rs61875193	C	0.202	0.084	0.088	0.087	0.110	0.010	0.085	0.011	0.108	0.016	0.083	0.002	0.010
rs73387413	A	0.266	0.126	0.135	0.124	0.062	0.170	0.120	0.074	0.073	0.199	0.127	0.078	0.086
rs12313615	A	0.298	0.152	0.165	0.158	0.400	0.260	0.150	0.180	0.380	0.268	0.150	0.169	0.191
rs117166327	A	0.177	0.069	0.071	0.085	-	-	0.060	.	0.000	0.000	0.066	0.000	0.000
rs62126348	C	0.186	0.073	0.077	0.071	0.043	0.020	0.100	0.075	0.046	0.037	0.077	0.093	0.075
rs11672710	T	0.186	0.073	0.076	0.071	0.036	0.019	0.100	0.075	0.042	0.035	0.077	0.093	0.075
rs62126347	A	0.186	0.073	0.077	0.069	0.023	0.019	0.100	0.075	0.027	0.034	0.077	0.093	0.075
rs10411896	C	0.186	0.074	0.078	0.072	0.460	0.230	0.100	0.320	0.445	0.220	0.081	0.291	0.326
rs1034726	C	0.189	0.069	0.069	0.065	-	-	-	-	-	-	-	-	-
rs12687449	T	0.226	0.094	0.095	0.093	-	-	0.058	-	0.000	0.000	0.088	0.000	0.000

^a^ Taiwan Precision Medicine Initiative. ^b^ Taiwan Biobank. ^c^ 1000 Genomes global minor allele frequency. ^d^ genom AD (genomes) allele frequencies. ^e^ AFR: African. AMR: American. EAS: East Asian. EUR: European. FIN: Finnish. NFE: Non-Finnish European.

**Table 4 diagnostics-14-00415-t004:** GO annotation of variant genes ^a^.

GO_ID	Term	*p*-Value	Gene
GO:0000291	nuclear-transcribed mRNA catabolic process, exonucleolytic	0.036	*LSM3*
GO:0001947	heart looping	0.044	*MICAL2*
GO:0003143	embryonic heart tube morphogenesis	0.048	*MICAL2*
GO:0006409	tRNA exports from the nucleus	0.035	*NUP210*
GO:0006662	glycerol-ether metabolic process	0.005	*TMEM86B*
GO:0007094	mitotic spindle assembly checkpoint signaling	0.022	*TEX14*
GO:0010921	regulation of phosphatase activity	0.029	*PPP6R1*
GO:0030042	actin filament depolymerization	0.011	*MICAL2*
GO:0031398	positive regulation of protein ubiquitination	0.003	*MAGEC2; HSPBP1*
GO:0033962	P-body assembly	0.013	*LSM3*
GO:0035304	regulation of protein dephosphorylation	0.042	*PPP6R1*
GO:0036151	phosphatidylcholine acyl-chain remodeling	0.031	*TMEM86B*
GO:0043268	positive regulation of potassium ion transport	0.039	*KCNN3*
GO:0043470	regulation of the carbohydrate catabolic process	0.044	*NUP210*
GO:0043666	regulation of phosphoprotein phosphatase activity	0.039	*PPP6R1*
GO:0043928	exonucleolytic catabolism of deadenylated mRNA	0.034	*LSM3*
GO:0045841	negative regulation of the mitotic metaphase/anaphase transition	0.023	*TEX14*
GO:0046485	ether-lipid metabolic process	0.015	*TMEM86B*
GO:0051031	tRNA transport	0.038	*NUP210*
GO:0051261	protein depolymerization	0.029	*MICAL2*
GO:0051438	regulation of ubiquitin-protein transferase activity	0.030	*MAGEC2*
GO:0051443	positive regulation of ubiquitin-protein transferase activity	0.032	*MAGEC2*
GO:0051547	regulation of keratinocyte migration	0.013	*IQSEC1*
GO:0051549	positive regulation of keratinocyte migration	0.010	*IQSEC1*
GO:0061371	determination of heart left/right asymmetry	0.048	*MICAL2*
GO:0071173	spindle assembly checkpoint signaling	0.022	*TEX14*
GO:0071174	mitotic spindle checkpoint signaling	0.022	*TEX14*
GO:0071431	tRNA-containing ribonucleoprotein complex exports from the nucleus	0.035	*NUP210*
GO:0097064	ncRNA exports from the nucleus	0.039	*NUP210*
GO:0120182	regulation of focal adhesion disassembly	0.006	*IQSEC1*
GO:0120183	positive regulation of focal adhesion disassembly	0.006	*IQSEC1*
GO:0150117	positive regulation of cell-substrate junction organization	0.025	*IQSEC1*
GO:1900542	regulation of the purine nucleotide metabolic process	0.044	*NUP210*
GO:1901381	positive regulation of potassium ion transmembrane transport	0.045	*KCNN3*
GO:1904064	positive regulation of cation transmembrane transport	0.042	*KCNN3*

^a^ Selected from *p*-value < 0.05.

**Table 5 diagnostics-14-00415-t005:** Reactome annotation of variant genes ^a^.

Reactome ID	Term	*p*-Value	Gene
R-HSA-72203	processing of capped intron-containing pre-mRNA	0.026	*NUP210*; *LSM3*
R-HSA-159227	transport of SLBP-independent mature mRNA	0.036	*NUP210*
R-HSA-159230	transport of SLBP-dependent mature mRNA	0.037	*NUP210*
R-HSA-159231	transport of mature mRNA derived from an intronless transcript	0.043	*NUP210*
R-HSA-159234	transport of mature mRNAs derived from intronless transcripts	0.044	*NUP210*
R-HSA-165054	Rev-mediated nuclear export of HIV RNA	0.036	*NUP210*
R-HSA-168271	transport of ribonucleoproteins into the host nucleus	0.036	*NUP210*
R-HSA-168274	export of viral ribonucleoproteins from the nucleus	0.037	*NUP210*
R-HSA-168276	NS1-mediated effects on host pathways	0.043	*NUP210*
R-HSA-168325	viral messenger RNA synthesis	0.048	*NUP210*
R-HSA-168333	NEP/NS2 interacts with cellular export machinery	0.036	*NUP210*
R-HSA-176033	interactions of Vpr with host cellular proteins	0.039	*NUP210*
R-HSA-177243	interactions of Rev with host cellular proteins	0.038	*NUP210*
R-HSA-180746	nuclear import of Rev protein	0.035	*NUP210*
R-HSA-180910	Vpr-mediated nuclear import of PICs	0.036	*NUP210*
R-HSA-430039	mRNA decays by 5 to 3 exoribonucleases	0.016	*LSM3*
R-HSA-1296052	Ca^2+^-activated K^+^ channels	0.008	*KCNN3*
R-HSA-1482788	acyl chain remodeling of PC	0.028	*TMEM86B*
R-HSA-3232142	SUMOylation of ubiquitinoylation proteins	0.040	*NUP210*
R-HSA-3301854	nuclear pore complex (NPC) disassembly	0.037	*NUP210*
R-HSA-4085377	SUMOylation of SUMOylation proteins	0.036	*NUP210*
R-HSA-4570464	SUMOylation of RNA-binding proteins	0.048	*NUP210*
R-HSA-4615885	SUMOylation of DNA replication proteins	0.046	*NUP210*
R-HSA-5619107	defective TPR may confer susceptibility to thyroid papillary carcinoma	0.033	*NUP210*
R-HSA-8875513	MET interacts with TNS proteins	0.005	*TNS3*
R-HSA-8875878	MET promotes cell motility	0.029	*TNS3*

^a^ Selected from *p*-value < 0.05.

**Table 6 diagnostics-14-00415-t006:** Disease association analysis from DisGeNET ^a^.

Gene	Immune Cell Type Expression Cluster ^a^	Immune Cell Type Specificity ^b^
*C12orf75*	Plasmacytoid DCs—Unknown function (mainly)	Immune cell enhanced (plasmacytoid DC)
*FAM209B*	Eosinophils—Unknown function (mainly)	Immune cell-enhanced (eosinophil)
*HSPBP1*	Non-specific—Mitochondria (mainly)	Low immune cell specificity
*IQSEC1*	Non-specific—Transcription (mainly)	Low immune cell specificity
*KCNN3*	Basophils—Unknown function (mainly)	Immune cell-enhanced (basophil and memory B-cell)
*LSM3*	Non-specific—Inflammatory response (mainly)	Low immune cell specificity
*MICAL2*	Neutrophils—Inflammatory response (mainly)	Immune cell-enhanced (neutrophil)
*NUP210*	Non-specific—mRNA processing (mainly)	Low immune cell specificity
*PPP6R1*	Plasmacytoid DCs—Unknown function (mainly)	Immune cell enhanced (plasmacytoid DC)
*SOCAR*	Basophils—Unknown function (mainly)	Low immune cell specificity
*SUSD5*	B-cells—Adaptive immune response (mainly)	Not detected in immune cells
*TMEM86B*	T-cells—Unknown function (mainly)	Not detected in immune cells
*TNS3*	Monocytes—Unknown function (mainly)	Group-enriched (classical monocyte, plasmacytoid DC, non-classical monocyte, intermediate monocyte, myeloid DC, memory B-cell, and naive B-cell)

^a^ RNA expression data has been used to classify protein-coding genes, and similar gene expression patterns were clustered together into immune cells. ^b^ Consensus transcriptomics data was used to classify all genes according to their immune cell-specific expression into two different schemas: the specificity category and the distribution category, based on the evidence data from the Human Protein Atlas.

## Data Availability

Data is contained within the article.

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
