# Peer review of "Host-Pathogen Interactions in K. pneumoniae Urinary Tract Infections: Investigating Genetic Risk Factors in the Taiwanese Population"

_diagnostics, 2024, doi:10.3390/diagnostics14040415_

Round 1

Reviewer 1 Report (Previous Reviewer 2)

Comments and Suggestions for Authors

The analyses of patient demography were improved in the revised manuscript, which show no statistically significant difference exists between cases and control patients. This study certainly proved valuable informations, however the reviewer still is uncertain whether this study would contribute to the diagnositics of infections diseases. It is up to the Editor whether to accept this manuscript in this journal.

Author Response

We sincerely appreciate the reviewer' s perspective and the suggestion. We have made minor revisions to our article based on the valuable feedback from other reviewers. We have strengthened the discussion on the severity of K. pneumoniae infections and its damage to patients and public health cares, enabling readers to better understand the importance of this issue. At the same time, we know that our study still has limitations and room for improvement. Therefore, we have provided explanations to readers regarding the potential limitations of our research in the discussion section. We hope that through these revisions, readers will be inspired to engage in more discussions and insights regarding this issue.

Reviewer 2 Report (Previous Reviewer 3)

Comments and Suggestions for Authors

The authors made the necessary corrections suggested by the reviewers and therefore the article is ready for publication

Author Response

We sincerely appreciate the reviewer' s perspective and the suggestion. We still have made some minor revisions to our article based on the valuable feedback from other reviewers. The most important addition is in the Discussion section, where we have included limitations of our study. We recognize that our research still has areas for improvement, and thus, we have provided explanations to readers regarding these limitations. We hope that through these revisions, readers will be inspired to engage in more discussions and gain further insights into the topic.

Reviewer 3 Report (New Reviewer)

Comments and Suggestions for Authors

Congratulation for your work and your efforts to support with your data the possibility that certain genome expression will reveal susceptible population for UTI development with K.pneumoniae. We always encourage investigators who orchestrate efforts and ideas to provide the scientific community with new and significant information. I personally acknowledge that you have contributed laboriously to define your objectives but I have some comments about the methodology followed and the interpretation of your results. Noteworthy the extensive analysis of your data provided valuable information but however there are some points that I would like to comment and your response will be definitely appreciated.

Comment 1: In the title the term UTI must be defined. Even more you need to specify a determination for what exactly the possible increased risk score is targeted.

Comment 2: Please inform us why the numbers (1), (2), (3), (4) are referred in the abstract body and for what element are indicative?

Comment 3: In the methodology and the limitation of the study must be commented the very low number of the Case Group specimen sample in order to name your investigative procedure as a GWAs. Even more the participants with increased morbidity are extremely few to make clear conclusions and it is not sensible to state that your results can be normalized to the presence of the very few cases for each comorbidity in the case group.

Comment 4: For the safe interpretation of your results, while you have located genes that are participating in the inflammation process, it would be crucial to know the relevance of your findings in the study population characterized with recurrence of the UTIs.

Comment 5: In the discussion section you should include known information about K.pneumonia variants with their clinical significance.

Comment 6: In the discussion section there is a paragraph for lung disease but no any information about the genitourinary tract infections.

Comment 7: You have to include study limitations and strengths in the discussion section.

Comments on the Quality of English Language

The use of English language must be improved.

Author Response

We appreciate the reviewer raising several thought-provoking points regarding our study conclusions and recommendations. Upon reflection of these critiques, we agree that certain aspects of the limitation of our research and we have expanded the Discussion section to provide possible limitation of our article to avoid causing misunderstanding.

Below are point by point about our responses to your comments

We genuinely appreciate this feedback to strengthen clarity and readability.

#Comment 1: In the title the term UTI must be defined. Even more you need to specify a determination for what exactly the possible increased risk score is targeted.
Ans: Thank you for pointing this out, and we have revised the phrase with urinary tract infection instead of UTI and the main goal of our research is to find out the possible genetic facotrs that make the host are more volunable to be infected by K.pneumonia in urinary tract.

#Comment 2: Please inform us why the numbers (1), (2), (3), (4) are referred in the abstract body and for what element are indicative?
Ans: Thank you for pointing this out, and we have revised the abstract and removed the numbers which was a misunderstainding and misspelling.

#Comment 3: In the methodology and the limitation of the study must be commented the very low number of the Case Group specimen sample in order to name your investigative procedure as a GWAs. Even more the participants with increased morbidity are extremely few to make clear conclusions and it is not sensible to state that your results can be normalized to the presence of the very few cases for each comorbidity in the case group.
Ans: Thank you for pointing this out and we have demonstrated the study limitation in the discussion sections (line 331) to notify the possible pitfall and shortage of our research.

#Comment 4: For the safe interpretation of your results, while you have located genes that are participating in the inflammation process, it would be crucial to know the relevance of your findings in the study population characterized with recurrence of the UTIs.
Ans: We sincerely appreciate the reviewer' s perspective and the suggestion. We do know UTI have high recurrent, but it is difficult to compare the recurrent rate and the prognosis between the case group and control group because it is difficult to define the first episode of UTI and track the following medical history of recurrent UTI events and further study are needed in order to illustrate the difference between these groups. However, we believe that it has little impact on affecting the significance of our results. 

#Comment 5: In the discussion section you should include known information about K.pneumonia variants with their clinical significance.
Ans: We sincerely appreciate the reviewer' s perspective and the suggestion. We have extended the content by including known information about K.pneumonia variants and their clinical significance (line 233). 

#Comment 6: In the discussion section there is a paragraph for lung disease but no any information about the genitourinary tract infections.
Ans: Thank you for pointing this out. The reason why we include this reference in our discussion, is that it had clearly illustrate the molecular pathogenesis of K.pneumonia infection. Although it is focus on lung disease, we believe that it may also share simialr pathway in invasion of urologic system.

#Comment 7: You have to include study limitations and strengths in the discussion section.
Ans: Thank you for pointing this out and we have deomstrate the study limitation in the discussion sections (line 331) to notify the possible pitfall and shortage of our research.

Reviewer 4 Report (New Reviewer)

Comments and Suggestions for Authors

The paper describes the associations of K. pneumoniae resistance with the SNPs in patients with Urinary tract infections. Taking into account the understanding of the impact of biofilms on bacterial susceptibility to antimicrobials,  this topic opens new inside in development of antibiofilm agents and is very important for the clinical practice.

The work scientifically sounds and can be considered for publication. Nevertheless, some issues should be adressed. Overall, while the experimental design is relevant, and idea is publication-worth, there are some minor issues to be adressed regarding data presentation.

Major

The introduction should be extended to description of K.pneumonia-driven deseases and outcomes. 

Minor

Please write Urinary tract infection instead of UTI in abstract

Use term association instead of connection

Comments on the Quality of English Language

The language should be checked throughout the manuscript (the style, doubling of verbs)

Author Response

We appreciate the reviewer raising several thought-provoking points regarding our study conclusions and recommendations. 

Below are point by point about our responses to your comments

We genuinely appreciate this feedback to strengthen clarity and readability.

Major:
The introduction should be extended to description of K.pneumonia-driven deseases and outcomes. 
Ans: We sincerely appreciate the reviewer’s perspective and the suggestion. We have extended the introduction about the threat caused by K.pneumonia-driven diseases to patient in hospital and nursing home (line56), and the notable  medical cost when dealing them (line58). 
Minor
Please write Urinary tract infection instead of UTI in abstract
Ans: Thank you for pointing this out, we have revised the phrase with urinary tract infection instead of UTI.

This manuscript is a resubmission of an earlier submission. The following is a list of the peer review reports and author responses from that submission.

Round 1

Reviewer 1 Report

Comments and Suggestions for Authors

The authors identified 11 genetic variants with higher odds ratios in patients with Klebsiella pneumoniae urinary tract infections (Kp UTI) compared to patients with other urinary tract infections (UTI). However, there are concerns about the interpretation of these findings:

1.     Similar Infection Mechanisms: The study suggests that the infection mechanisms of Kp UTIs and Escherichia coli (Ec) UTIs are very similar. Therefore, the detected variants may be reflective of population differences rather than specific to K. pneumoniae UTIs.

2.     Variant Location: The detected variants are located in either intergenic or intronic regions, which may indicate that they have no functional relevance. Functional annotation is crucial to understanding the potential impact of these variants.

3.     Medical History Comparison: The study did not compare the medical history between the case and control groups, and there may be other feature differences beyond UTI pathogen distinctions.

The authors are advised to verify these variants in other cohorts to strengthen the validity and generalizability of their study. Additionally, conducting functional studies on the identified variants and considering relevant clinical parameters in the analysis would enhance the robustness of the findings.

Comments on the Quality of English Language

The English is OK.

Reviewer 2 Report

Comments and Suggestions for Authors

Comments to the authors:

1. The reviewer acknowledges the efforts the authors have paid for this study, and the genes the authors have identified could have certain role in the pathogenesis of UTIs by Klebsiella. However, this study is purely focused on the genetic aspect of UTIs by Klebsiella, and the genetic information was not related to clinical manifestations. From this point, it is hard to say that this study is focused on the diagnostic microbiology, and seems to be unsuitable to be submitted in this journal. The reviewer suggest the authors to submitt this manuscript to more suitable journal.

2. in Table 1, medical histories were described for case groups 1 and 2, but not for control group. Is there any reason for this lack of information? If there would be any difference in the medical histories between case group and control group, and if the difference in medical histories would be ralated to the expression of specific genes, the authors's discoveries would be more appreciated in the aspect of clinical microbiology.

Comments on the Quality of English Language

Minor grammatical errows were identified throughout the manuscript.

Reviewer 3 Report

Comments and Suggestions for Authors

The present work contributes significantly to the medical literature. Addresses urinary tract infection in a country with a high incidence of muti-resistant strains of Klebsiella pneumoniae. It presents a genomic study that correlates detected genes with clinical manifestations and risk of death. The language is fine and the references are in accordance with the proposal presented. Therefore, I recommend the work for publication in the magazine.